# Is RobustBench/AutoAttack a suitable Benchmark for Adversarial Robustness?

**Peter Lorenz**[1,2,3], **Dominik Straßel**[1,2], **Margret Keuper**[4] **and Janis Keuper**[1,2,5]

[1] Competence Center High Performance Computing, Fraunhofer ITWM, Kaiserslautern, Germany
[2] Fraunhofer Research Center Machine Learning, Germany
[3] Computer Vision and Learning Lab, Heidelberg University, Germany
[4] University of Siegen, Max Planck Institute for Informatics, Saarland Informatics Campus, Germany
[5] Institute for Machine Learning and Analytics (IMLA), Offenburg University, Germany
Correspondence to peter.lorenz@itwm.fhg.de

## Abstract

Recently, *RobustBench* (Croce et al. 2020) has become a widely recognized benchmark for the adversarial robustness of image classification networks. In its most commonly reported sub-task, *RobustBench* evaluates and ranks the adversarial robustness of trained neural networks on *CIFAR10* under AutoAttack (Croce and Hein 2020b) with $l_\infty$ perturbations limited to $\epsilon = 8/255$. With leading scores of the currently best performing models of around $60\%$ of the baseline, it is fair to characterize this benchmark to be challenging.

Despite its general acceptance in recent literature, we aim to foster discussion about the suitability of *RobustBench* as a key indicator for robustness which could be generalized to practical applications. Our line of argumentation against this is two-fold and supported by excessive experiments presented in this paper: We argue that I) the alternation of data by AutoAttack with $l_\infty, \epsilon = 8/255$ is unrealistically strong, resulting in close to perfect detection rates of adversarial samples even by simple detection algorithms while other attack methods are much harder to detect and achieve similar success rates, II) results on low resolution data sets like CIFAR10 do not generalize well to higher resolution images as gradient based attacks appear to become even more detectable with increasing resolutions.

Source code: github.com/adverML/SpectralDef_Framework

## 1 Introduction

Increasing the robustness of neural network architectures against adversarial examples in general and more specifically against coordinated adversarial attacks, has recently received increasing attention. In this work, we focus on the benchmarking of robustness in the context of CNN based computer vision models.

**RobustBench.** In 2020, (Croce et al. 2020) launched a benchmark website[1] with the goal to provide a standardized benchmark for adversarial robustness on image classification models. Until then, single related libraries such as Fool-Box (Rauber, Brendel, and Bethge 2018), Cleverhans (Papernot et al. 2016) and AdverTorch (Ding, Wang, and Jin

[1]robustbench.github.io

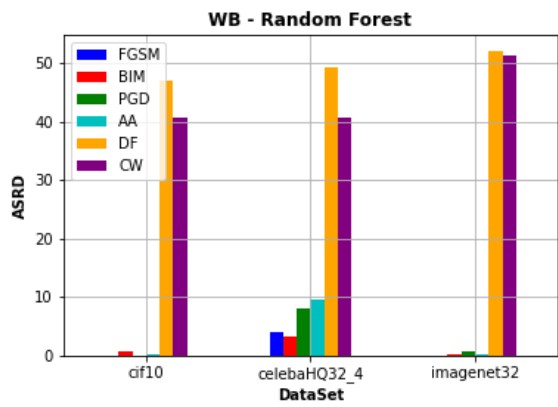

Figure 1: Attack Success Rates under Defence (ASRD) of different adversarial attack methods on several datasets for a simple defense:White-Box Fourier domain detector with random forest (Harder et al. 2021): *RobustBench's AutoAttack* are so easy to detect that successful attacks are very unlikely compared with other methods.

2019) were already available but did not include all state-of-the-art (SOTA) methods in one evaluation.

The current rankings in *RobustBench* as well as the majority of evaluations of adversarial robustness in recent literature are dominated by *RobustBench's* own attack scheme *AutoAttack* (Croce and Hein 2020b). *AutoAttack* is an ensemble of 4 attacks: two variations of the Projected Gradient Descent (PGD) (Madry et al. 2018) attack with cross-entropy loss (APGD-CE) and difference of logits ratio loss (APGD-t), the targeted version of the FAB attack (Croce and Hein 2020a), and the black-box Squares attack (Andriushchenko et al. 2020).

**Contributions** The aim of this paper is to raise the awareness that *RobustBench's AutoAttack* in its default evaluation scheme $l_\infty, \epsilon = 8/255$ is unrealistically strong, resulting in close to perfect detection rates of adversarial samples even by simple detection algorithms. Also we find that benchmarks on low resolution datasets like CIFAR10 tend to underestimate the strength of adversarial attacks and can not directly generalized to applications with higher resolutions.

In detail, we show that:

- adversarial samples generated by *AutoAttack* $l_\infty, \epsilon = 8/255$ are modifying test images to the extent that these manipulations can easily be detected, almost entirely preventing successful attacks in practice.
- given a simple defense, *AutoAttack* is outperformed by other existing attacks even for optimized $\epsilon$ parameters.
- in contrast to other methods, the effectiveness of *AutoAttack* is dropping with increasing image resolutions.

## 2 Methods

### 2.1 Attack Methods

For our analysis, we generate test data using *AutoAttack* and a baseline of five other commonly used attack methods from the *foolbox* (Rauber, Brendel, and Bethge 2018). We employ the untargetet version of all attacks, if available.

**AutoAttack (AA):** *RobustBench* is based on the evaluation of AA (Croce and Hein 2020b), which is an ensemble of 4 parameter-free attacks: two variations of the PGD attack (Madry et al. 2018) (see Section 2.1) with cross-entropy loss (APGD-CE) and difference of logits ratio loss (APGD-t):

$$\text{DLR}(x, y) = \frac{z_y - \max_{x \neq y} z_i}{z_{\pi 1} - z_{\pi 3}}.$$

where $\pi$ is ordering of the components of $z$ in decreasing order. The APGD-t can handle models with a minimum of 4 classes. The targeted version of the FAB attack (Croce and Hein 2020a), and the Black-Box (BB) Squares attack (Andriushchenko et al. 2020). The AA framework provides two modes. *RobustBench* uses the "standard" mode, executing the 4 attack methods consecutively. The failed attacked samples are handed over to the next attack method, to ensure an higher attack rate.

**Fast Gradient Method (FGSM):** The FGSM (Goodfellow, Shlens, and Szegedy 2015) uses the gradients of the Deep Neural Network (DNN) to create adversarial examples. For an input image, the method uses the gradients of the loss w.r.t. the input image to create a new image that maximises the loss. This output is called the adversarial image. This following expression summarizes this:

$$X^{adv} = X - \varepsilon \text{sign}(\nabla_X J(X_N^{adv}, y_t)),$$

where $X^{adv}$ is the adversarial image, $X$ is the original input image, $y$ is the original input label, $\varepsilon$ is the multiplier to ensure the perturbations are small and $J$ is the loss. There is no guarantee that the generated adversarial examples by this method are similar to its real counterpart.

**Basic Iterative Method (BIM):** The method BIM (Kurakin, Goodfellow, and Bengio 2017) is the iterative version of FGSM. After each iteration the pixel values need to be clipped to ensure the generated adversarial examples is still within the range of both the $\varepsilon$ ball (i.e. $[x - \varepsilon, x + \varepsilon]$) and the input space (i.e. $[0, 255]$ for the pixel values). The formulation is expressed as follows:

$$X_0^{adv} = X,$$
$$X_{N+1}^{adv} = \text{CLIP}_{X,\varepsilon} \{ X_N^{adv} - \alpha \text{sign}(\nabla_X J(X, y_t)) \},$$

where $N$ denotes the number of iterations.

**Projected Gradient Descent (PGD):** The PGD (Madry et al. 2018) is a variant of BIM and one of the most popular white-box (allowing full access to model gradients and weights) attacks. It introduces random initialization of the perturbations for each iteration. This algorithm strives to find the perturbation that maximizes a model's loss on a particular input. The size of the perturbation is kept smaller than an amount by $\epsilon$. This constraint is expressed ether as $l_2$ or $l_\infty$ norm.

**DeepFool (DF):** The DF is a non-targed method that is able to find the minimal amount of perturbations possible which mislead the model using an iterative linearization approach (Moosavi-Dezfooli, Fawzi, and Frossard 2016). The main idea is to find the closest distance from the input sample to the model decision boundary.

**Carlini&Wagner (C&W):** The attack method Carlini&Wagner (C&W) (Carlini and Wagner 2017) is based on the L-BFGS and has three versions: $l_0$, $l_2$ and $l_\infty$. We employ the $l^2$ variant which is most commonly used. This attack method generates for an given input $X$ an adversarial example $X^{adv}$ by formulating following optimization problem:

$$\min \| \frac{1}{2}(\tanh(X^{adv}) + 1) - X \| + cf(\frac{1}{2}(\tanh(X^{adv}) + 1))$$
$$\text{With } f(x) = \max(Z(x)_{true} - \max_{i \neq true}\{Z(x)_i\}, 0),$$

where $Z(x)$ is the softmax classification result vector. The initial value for $c$ is $c = 10^{-3}$, a binary search is performed to then find the smallest $c$, s.t. $f(X_{adv}) \leq 0$.

### 2.2 Measuring the Success of Adversarial Attacks

*RobustBench*, like most of the benchmarks in literature regarding adversarial robustness, uses a *Robust Accuracy* (Croce et al. 2020) measure to compare different methods. However, this approach does not fit our evaluation scheme, since we are aiming to measure the success of adversarial samples under defence in order to obtain a more realistic view on the practical impact of the applied attacks. Therefore, we reformulate the robustness measures and report two different indicators:

**Attack Success Rate (ASR)** The *Adversarial Succes Rate (ASR)* in eq. (1) is calculated as

$$\text{ASR} = \frac{\text{\# perturbed samples}}{\text{\# all samples}} \tag{1}$$

the fraction of successfully perturbed test images and it provides a baseline of an attacker's ability to fool unprotected target networks. Hence, *ASR* is providing the same information as *Robust Accuracy* from an attackers perspective.

**Attack Success Rate under Defense (ASRD)** We extend *ASR* by the practical assumption that too strong perturbations can be detected at inference time. To measure the performance of attacks under defense, we introduce the *Adversarial Success Rate under Detection (ASRD)* in eq. (2), computing the ratio of successful attacks

$$\text{ASRD} = \frac{\text{\# undetected perturbations}}{\text{\# all samples}} = \text{FNR} \cdot \text{ASR}, \tag{2}$$

where FNR is the false negative rate of the applied detection algorithm.

## 2.3 A Simple Adversarial Detector

In order to measure the magnitude of perturbations imposed by *RobustBench*, we apply a simple and easy to implement adversarial detector introduced in (Harder et al. 2021; Lorenz et al. 2021). This method is based on a feature extraction in the Fourier domain, followed by a *Logistic Regression* or *Random Forest* classifier. It can be applied in a black-box fashion, using only the (adversarial) input images, or as white-box detector accessing the feature maps of attacked neural networks. In both cases, the detector is based on a Fourier transformation (Cooley and Tukey 1965): For a discrete 2D signal, like color image channels or single CNN feature maps − $X \in [0,1]^{N \times N}$ − the 2D discrete Fourier transform is given as

$$\mathcal{F}(X)(l,k) = \sum_{n,m=0}^{N} e^{-2\pi i \frac{lm+kn}{N}} X(m,n), \qquad (3)$$

for $l, k = 0, \ldots N-1$, with complex valued Fourier coefficients $\mathcal{F}(X)(l,k)$. The detector then only utilizes the magnitudes of Fourier coefficients

$$|\mathcal{F}(X)(l,k)| = \sqrt{\mathrm{Re}(\mathcal{F}(X)(l,k))^2 + \mathrm{Im}(\mathcal{F}(X)(l,k))^2} \qquad (4)$$

to detect adversarial attacks with high accuracy.

**Black-Box Detection: Fourier Features of Input Images**
While different attacks show distinct but randomly located change patterns in the spatial domain (which makes them hard to detect), (Harder et al. 2021) showed that adversarial samples have strong, well localized signals in the frequency domain.
Hence, the detector extracts and concatenates the 2D power spectrum of each color channel as feature representations of input images and uses simple classifiers like *Random Forests* and *Logistic Regression* to learn to detect perturbed input images.

**White-Box Detection: Fourier Features of Feature-Maps**
In the White-Boxcase, the detector applies the same method as in the Black-Boxapproach, but extends the inputs to the feature map responses of the target network to test samples. Since this extension will drastically increase the feature space for larger target networks, only a subsets of the available feature maps are selected. In original paper (Harder et al. 2021) and in the follow-up paper (Lorenz et al. 2021), it is stated that a combination of several layers delivers better detection results.

## 3 Experiments

Since most of the successful methods ranked on *Robustbench* are based on a WideResNet28-10 (Zagoruyko and Komodakis 2017) architecture, we also conduct our evaluation on a baseline WideResNet28-10 using the following datasets without applying adversarial examples or other methods to increase the robustness during training.

**CIFAR10.** We train on the plain CIFAR10 training set to a test-accuracy of 87% and apply the different attacks on the test set. Then, we extract the spectral features and use a random subset of 1500 samples of this data for each attack method to evaluate *ASR* and *ASRD* .

**CIFAR100.** The procedure is similar to CIFAR10 dataset. We train on the CIFAR100 training set to a test-accuracy of 79% and apply the attacks on the test set.

**ImageNet-32. (64 and 128.)** This dataset (Chrabaszcz, Loshchilov, and Hutter 2017) (and its variants $64 \times 64$ and $128 \times 128$ pixels) has the exact same number of classes (1000) and images as the original ImageNet with the only difference that the images are downsampled. Moreover, a lower resolution of the images makes the classification task more difficult and the baseline test accuracy is 66% and 77% respectively.

**CelebaHQ-32. (64 and 128.)** This dataset (Liu et al. 2015) provides images of celebrities faces in HQ quality ($1024 \times 1024px$) whereas we downsampled it to 32, 64 and 128 pixels width and height. We only selected the attributes "Brown Hair", "Blonde Hair", "Black Hair" and "Gray Hair" to train the Wide Residual Networks (WRN) to an test-accuracy of 91%. The data is unbalanced, where the class "Gray Hair" has least samples.

### 3.1 Detecting Attacks

Figures 1 and 2 show a subset of white-box and black-box ASRD results for all attack methods on datasets with a resolution of $32 \times 32$[2]. In both cases, *AutoAttack* has very low ASRD rates, not only compared to other methods but also in absolute values. In most cases, the probability of successful AA attacks is marginally low.

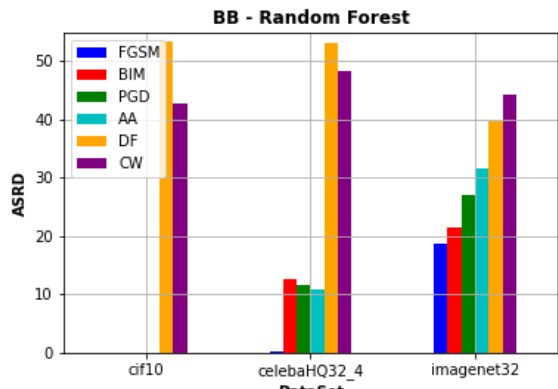

Figure 2: Black-box ASRD comparison using a Random Forest classifier on different $32 \times 32$ datasets.

---

[2]The full ASRD evaluation on all datasets is listed in table table 1 of the appendix.

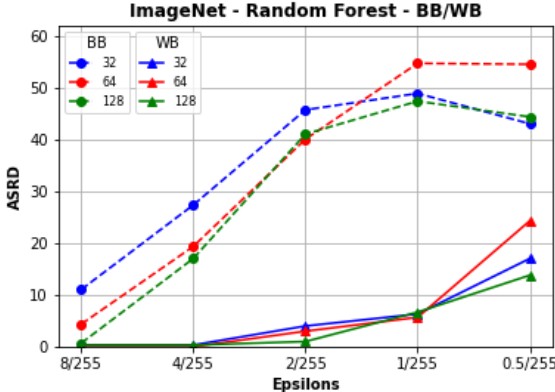

Figure 3: ASRD of AA with random forest for a range of different $\epsilon$ on ImageNet.

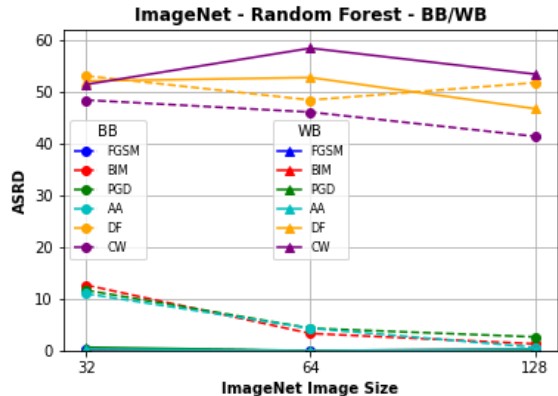

Figure 5: ASRD with Random Forest classifiers on increasing resolutions of ImageNet.

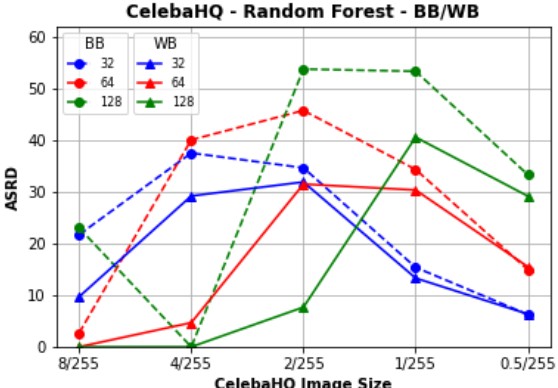

Figure 4: ASRD of AA with random forest for a range of different $\epsilon$ on CelebHQ.

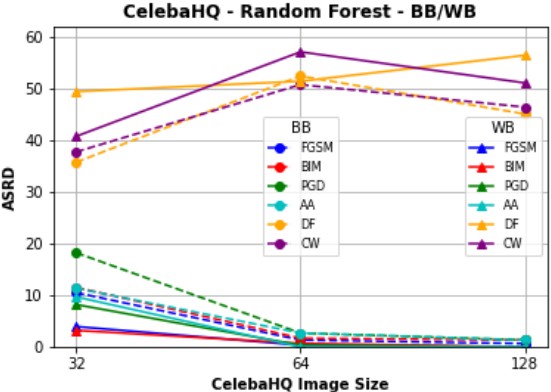

Figure 6: ASRD with Random Forest classifiers on increasing resolutions of CelebaHQ_4.

## 3.2 AutoAttack for different choices of $\epsilon$

One might argue that the low ASRD rates of AA might be caused by too high choice of $\epsilon$. Hence, we repeat the full set of *AutoAttack* experiments for a full range of different $\epsilon$-values. Figures 3 and 4 show a subset of these evaluation for ImageNet and CelebHQ on different $\epsilon$, image resolutions as well as WB and BB detectors with Random Forests[3].

## 3.3 Success Rates depending on Image Resolution

As shown in Figure 5 and 6, we compare the ASRD over the three image size ($s = \{32, 64, 128\}$) on the datasets CelebaHQ and ImageNet. The attacks FGSM, BIM, PGD and AA are sensitive to the image size. The used detector has better results as the image size is increased. In contrast, DF and C&W keep their attack strength over all image sizes $s$. Again, AA does not show sufficient results for using adversarial detection robustness.

## 4   Discussion

The results of our empirical evaluations show strong evidence that the widely used *AutoAttack* scheme for benchmarking the adversarial robustness of image classifier models on low resolution data might not be a suitable setup in order to generalize the obtained results to estimate the robustness in practical vision applications. Even for lower choices of the $\varepsilon$-parameter, *AutoAttack* still appears to modify target images beyond reasonable class boundaries. Additionally, the resolution of the benchmark images should not be neglected. In terms of resolution as well as in the number of classes and training images, CIFAR10 is a conveniently sized dataset for the very expensive state of the art adversarial training approaches. However, our experiments suggest that these results might not generalize to more complex problems.

In light of our results, we argue that too strong adversarial benchmarks like the current setting of *RobustBench* might hamper the development of otherwise practically relevant methods towards more model robustness.

---

[3]Full evaluation results in table Table 2 of the appendix.

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

| Arch: Wide ResNet 28-10 | | ASR | BB | | | | | | WB | | | | | |
|---|---|---|---|---|---|---|---|---|---|---|---|---|---|---|
| | | | F1 | | FNR | | ASRD | | F1 | | FNR | | ASRD | |
| | | | LR | RF | LR | RF | LR | RF | LR | RF | LR | RF | LR | RF |
| **Cif10** | FGSM | 95.08 | 97.34 | 97.72 | 2.33 | 0.00 | 2.22 | 0.00 | 99.01 | 97.88 | 0.00 | 0.00 | 0.00 | 0.00 |
| | BIM | 99.37 | 92.93 | 95.54 | 8.00 | 0.00 | 7.95 | 0.00 | 97.65 | 96.44 | 3.00 | 0.67 | 2.98 | 0.67 |
| | PGD | 99.27 | 91.79 | 95.24 | 8.67 | 0.00 | 8.61 | 0.00 | 96.70 | 95.85 | 2.33 | 0.00 | 2.31 | 0.00 |
| | AA | 100.0 | 91.78 | 96.31 | 7.00 | 0.00 | 7.00 | 0.00 | 98.00 | 96.76 | 2.00 | 0.33 | 2.00 | 0.33 |
| | DF | 100.0 | 48.31 | 49.47 | 54.67 | 53.33 | 54.67 | 53.33 | 54.42 | 52.30 | 45.67 | 47.00 | 45.67 | 47.00 |
| | CW | 100.0 | 48.07 | 53.75 | 54.33 | 42.67 | 54.33 | 42.67 | 53.29 | 54.52 | 47.33 | 40.67 | 47.33 | 40.67 |
| **Cif100** | FGSM | 99.95 | 94.58 | 97.72 | 7.00 | 0.00 | 7.00 | 0.00 | 99.34 | 98.85 | 0.33 | 0.00 | 0.33 | 0.00 |
| | BIM | 99.95 | 87.39 | 95.39 | 15.67 | 0.00 | 15.66 | 0.00 | 97.00 | 98.50 | 3.00 | 1.33 | 3.00 | 1.33 |
| | PGD | 99.95 | 86.97 | 95.24 | 14.33 | 0.00 | 14.32 | 0.00 | 96.83 | 98.68 | 3.33 | 0.00 | 3.33 | 0.00 |
| | AA | 100.0 | 92.57 | 96.76 | 8.67 | 0.33 | 8.67 | 0.33 | 97.35 | 97.72 | 2.00 | 0.00 | 2.00 | 0.00 |
| | DeepFool | 100.0 | 50.17 | 51.84 | 49.67 | 46.00 | 49.67 | 46.00 | 50.33 | 48.00 | 49.33 | 54.00 | 49.33 | 54.00 |
| | CW | 100.0 | 50.17 | 64.20 | 49.67 | 10.33 | 49.67 | 10.33 | 47.92 | 47.29 | 54.00 | 55.00 | 54.00 | 55.00 |
| **ImageNet32** | FGSM | 99.95 | 84.53 | 90.20 | 15.33 | 0.33 | 15.32 | 0.33 | 100.0 | 99.83 | 0.00 | 0.00 | 0.00 | 0.00 |
| | BIM | 100.0 | 71.33 | 78.68 | 30.33 | 12.67 | 30.33 | 12.67 | 100.0 | 99.67 | 0.00 | 0.33 | 0.00 | 0.33 |
| | PGD | 100.0 | 74.70 | 78.75 | 26.67 | 11.67 | 26.67 | 11.67 | 100.0 | 99.67 | 0.00 | 0.67 | 0.00 | 0.67 |
| | AA | 100.0 | 71.74 | 79.82 | 29.33 | 11.00 | 29.33 | 11.00 | 99.67 | 99.67 | 0.00 | 0.33 | 0.00 | 0.33 |
| | DeepFool | 100.0 | 66.59 | 48.45 | 0.33 | 53.00 | * | 53.00 | 50.33 | 48.98 | 49.33 | 52.00 | 49.33 | 52.00 |
| | CW | 100.0 | 66.59 | 50.82 | 0.33 | 48.33 | * | 48.33 | 51.46 | 49.41 | 47.00 | 51.33 | 47.00 | 51.33 |
| **ImageNet64** | FGSM | 100.0 | 88.15 | 92.59 | 12.00 | 0.00 | 12.00 | 0.00 | 99.83 | 99.67 | 0.00 | 0.00 | 0.00 | 0.00 |
| | BIM | 100.0 | 74.29 | 84.30 | 26.33 | 3.33 | 26.33 | 3.33 | 99.50 | 99.17 | 0.33 | 0.00 | 0.33 | 0.00 |
| | PGD | 100.0 | 75.63 | 82.59 | 25.00 | 4.33 | 25.00 | 4.33 | 99.67 | 99.67 | 0.33 | 0.00 | 0.33 | 0.00 |
| | AA | 100.0 | 78.54 | 81.42 | 21.33 | 4.33 | 21.33 | 4.33 | 99.83 | 99.67 | 0.00 | 0.00 | 0.00 | 0.00 |
| | DeepFool | 100.0 | 49.32 | 50.82 | 51.33 | 48.33 | 51.33 | 48.33 | 50.66 | 48.63 | 48.67 | 52.67 | 48.67 | 52.67 |
| | CW | 100.0 | 60.84 | 51.92 | 22.33 | 46.00 | * | 46.00 | 49.24 | 45.29 | 51.67 | 58.33 | 51.67 | 58.33 |
| **ImageNet128** | FGSM | 100.0 | 89.55 | 92.88 | 10.00 | 0.00 | 10.00 | 0.00 | 99.83 | 99.34 | 0.00 | 0.00 | 0.00 | 0.00 |
| | BIM | 100.0 | 81.43 | 91.36 | 20.33 | 1.33 | 20.33 | 1.33 | 99.50 | 98.52 | 0.00 | 0.33 | 0.00 | 0.33 |
| | PGD | 100.0 | 81.82 | 90.82 | 19.00 | 2.67 | 19.00 | 2.67 | 99.67 | 99.34 | 0.00 | 0.00 | 0.00 | 0.00 |
| | AA | 100.0 | 77.34 | 85.51 | 18.67 | 0.67 | 18.67 | 0.67 | 99.34 | 98.19 | 0.00 | 0.33 | 0.00 | 0.33 |
| | DeepFool | 100.0 | 66.67 | 49.15 | 0.00 | 51.67 | * | 51.67 | 53.85 | 51.61 | 41.67 | 46.67 | 41.67 | 46.67 |
| | CW | 100.0 | 60.00 | 53.99 | 25.00 | 41.33 | * | 41.33 | 54.41 | 48.19 | 40.33 | 53.33 | 40.33 | 53.33 |
| **CelebaHQ32_4** | FGSM | 78.59 | 75.95 | 76.64 | 23.67 | 18.00 | 18.60 | 14.15 | 85.95 | 93.44 | 13.33 | 5.00 | 10.48 | 3.93 |
| | BIM | 95.91 | 73.97 | 74.06 | 22.33 | 21.00 | 21.42 | 20.14 | 84.48 | 96.35 | 12.00 | 3.33 | 11.51 | 3.19 |
| | PGD | 90.93 | 71.40 | 68.99 | 29.67 | 30.67 | 26.98 | 27.89 | 79.47 | 91.46 | 20.00 | 9.00 | 18.19 | 8.18 |
| | AA | 100.0 | 69.49 | 74.25 | 31.67 | 21.67 | 31.67 | 21.67 | 87.79 | 88.71 | 11.33 | 9.67 | 11.33 | 9.67 |
| | DeepFool | 100.0 | 59.05 | 49.32 | 39.67 | 52.00 | 39.67 | 52.00 | 63.59 | 57.69 | 35.67 | 49.33 | 35.67 | 49.33 |
| | CW | 100.0 | 55.76 | 48.64 | 44.33 | 52.33 | 44.33 | 52.33 | 61.11 | 58.46 | 37.67 | 40.67 | 37.67 | 40.67 |
| **CelebaHQ64_4** | FGSM | 100.0 | 93.27 | 90.97 | 5.33 | 4.33 | 5.33 | 4.33 | 98.01 | 99.67 | 1.33 | 0.33 | 1.33 | 0.33 |
| | BIM | 100.0 | 95.16 | 95.30 | 5.00 | 2.00 | 5.00 | 2.00 | 98.66 | 99.50 | 1.67 | 0.67 | 1.67 | 0.67 |
| | PGD | 100.0 | 90.85 | 91.67 | 9.00 | 4.67 | 9.00 | 4.67 | 97.17 | 99.50 | 2.67 | 0.33 | 2.67 | 0.33 |
| | AA | 100.0 | 84.26 | 84.60 | 14.33 | 5.67 | 14.33 | 5.67 | 97.17 | 100.0 | 2.67 | 0.00 | 2.67 | 0.00 |
| | DeepFool | 100.0 | 48.08 | 47.04 | 54.00 | 55.00 | 54.00 | 55.00 | 49.31 | 49.66 | 52.33 | 51.33 | 52.33 | 51.33 |
| | CW | 100.0 | 50.25 | 50.89 | 50.00 | 47.33 | 50.00 | 47.33 | 50.25 | 45.58 | 50.67 | 57.00 | 50.67 | 57.00 |
| **CelebaHQ128_4** | FGSM | 95.74 | 98.82 | 97.40 | 2.00 | 0.00 | 1.91 | 0.00 | 99.67 | 100.0 | 0.67 | 0.00 | 0.64 | 0.00 |
| | BIM | 99.95 | 98.16 | 98.03 | 2.00 | 0.33 | 2.00 | 0.33 | 99.16 | 100.0 | 1.33 | 0.00 | 1.33 | 0.00 |
| | PGD | 99.76 | 97.37 | 98.20 | 1.33 | 0.00 | 1.33 | 0.00 | 99.16 | 100.0 | 1.33 | 0.00 | 1.33 | 0.00 |
| | AA | 100.0 | 93.57 | 92.88 | 3.00 | 0.00 | 3.00 | 0.00 | 98.67 | 100.0 | 1.33 | 0.00 | 1.33 | 0.00 |
| | DeepFool | 100.0 | 55.21 | 52.98 | 44.33 | 46.67 | 44.33 | 46.67 | 55.65 | 50.87 | 45.00 | 56.33 | 45.00 | 56.33 |
| | CW | 100.0 | 51.63 | 50.50 | 47.33 | 49.00 | 47.33 | 49.00 | 52.87 | 50.26 | 46.33 | 51.00 | 46.33 | 51.00 |

**Appendix** 1: Results of the proposed detectors on AutoAttack (standard mode) for different choices of the hyper-parameter $\varepsilon$ (default in most publications is $\varepsilon = 8/255$) and test sets. ASR=Attack Success Rate, ASRD=Attack Success Rate under Detection. Black-Box (BB) and White-Box (WB) results on all datasets are obtained by a Logistic Regression classifier and Random Forests. F1 and the False Negative Rate (FNR) are used to report the detection performance. See Section 3 for details of the experimental setup. Note that ASRD values marked by a star '*' are missing values.

| Arch: Wide ResNet 28-10 | | ASR | BB | | | | | | WB | | | | | |
|---|---|---|---|---|---|---|---|---|---|---|---|---|---|---|
| | | | F1 | | FNR | | ASRD | | F1 | | FNR | | ASRD | |
| | | | LR | RF | LR | RF | LR | RF | LR | RF | LR | RF | LR | RF |
| **Cif10** | AA (8/255) | 100.0 | 91.78 | 96.31 | 7.00 | 0.00 | 7.00 | 0.00 | 98.00 | 96.76 | 2.00 | 0.33 | 2.00 | 0.33 |
| | AA (4/255) | 100.0 | 83.36 | 92.28 | 15.67 | 0.33 | 15.67 | 0.33 | 91.00 | 88.75 | 7.33 | 2.67 | 7.33 | 2.67 |
| | AA (2/255) | 94.41 | 69.26 | 82.39 | 31.67 | 10.33 | 29.90 | 9.75 | 83.63 | 79.00 | 14.00 | 16.00 | 13.22 | 15.11 |
| | AA (1/255) | 56.39 | 57.93 | 69.61 | 44.00 | 26.33 | 24.81 | 14.85 | 69.32 | 62.79 | 30.33 | 33.33 | 17.10 | 18.79 |
| | AA (0.5/255) | 23.14 | 52.67 | 41.33 | 55.52 | 10.95 | 47.33 | 9.56 | 58.55 | 50.00 | 40.67 | 51.00 | 9.41 | 11.80 |
| **Cif100** | AA (8/255) | 100.0 | 92.57 | 96.76 | 8.67 | 0.33 | 8.67 | 0.33 | 97.35 | 97.72 | 2.00 | 0.00 | 2.00 | 0.00 |
| | AA (4/255) | 99.90 | 83.93 | 91.93 | 17.33 | 1.33 | 17.31 | 1.33 | 91.61 | 92.11 | 9.00 | 4.67 | 8.99 | 4.67 |
| | AA (2/255) | 97.28 | 72.03 | 82.30 | 31.33 | 9.33 | 30.48 | 9.08 | 83.22 | 83.81 | 15.67 | 12.00 | 15.24 | 11.67 |
| | AA (1/255) | 73.65 | 62.81 | 70.77 | 36.67 | 23.33 | 27.01 | 17.18 | 73.89 | 74.04 | 25.00 | 19.67 | 18.41 | 14.49 |
| | AA (0.5/255) | 38.97 | 51.23 | 60.44 | 51.33 | 36.33 | 20.00 | 14.16 | 61.59 | 60.87 | 39.33 | 37.00 | 15.33 | 14.42 |
| **ImageNet32** | AA (8/255) | 100.0 | 71.74 | 79.82 | 29.33 | 11.00 | 29.33 | 11.00 | 99.67 | 99.67 | 0.00 | 0.33 | 0.00 | 0.33 |
| | AA (4/255) | 99.95 | 62.38 | 65.27 | 37.00 | 27.33 | 36.98 | 27.32 | 99.00 | 97.71 | 0.67 | 0.33 | 0.67 | 0.33 |
| | AA (2/255) | 100.0 | 56.58 | 55.54 | 42.67 | 45.67 | 42.67 | 45.67 | 96.82 | 94.27 | 3.67 | 4.00 | 3.67 | 4.00 |
| | AA (1/255) | 99.67 | 51.82 | 50.33 | 47.67 | 49.00 | 47.51 | 48.84 | 87.67 | 89.21 | 12.33 | 6.33 | 12.29 | 6.31 |
| | AA (0.5/255) | 92.78 | 52.55 | 51.60 | 45.00 | 46.33 | 41.75 | 42.98 | 79.47 | 76.56 | 20.00 | 18.33 | 18.56 | 17.01 |
| **ImageNet64** | AA (8/255) | 100.0 | 78.54 | 81.42 | 21.33 | 4.33 | 21.33 | 4.33 | 99.83 | 99.67 | 0.00 | 0.00 | 0.00 | 0.00 |
| | AA (4/255) | 100.0 | 65.37 | 72.56 | 33.00 | 19.33 | 33.00 | 19.33 | 99.00 | 99.01 | 1.33 | 0.00 | 1.33 | 0.00 |
| | AA (2/255) | 100.0 | 58.84 | 58.06 | 39.00 | 40.00 | 39.00 | 40.00 | 97.03 | 94.02 | 2.00 | 3.00 | 2.00 | 3.00 |
| | AA (1/255) | 99.95 | 50.53 | 47.47 | 52.00 | 54.67 | 51.97 | 54.64 | 88.36 | 89.70 | 12.67 | 5.67 | 12.66 | 5.67 |
| | AA (0.5/255) | 98.40 | 48.06 | 46.37 | 54.67 | 55.33 | 53.80 | 54.44 | 67.38 | 71.97 | 37.00 | 24.67 | 36.41 | 24.28 |
| **ImageNet128** | AA (8/255) | 100.0 | 77.34 | 85.51 | 18.67 | 18.67 | 18.67 | 0.67 | 99.34 | 98.19 | 0.00 | 0.33 | 0.00 | 0.33 |
| | AA (4/255) | 100.0 | 59.97 | 72.38 | 42.33 | 42.33 | 42.33 | 17.00 | 97.52 | 96.61 | 1.67 | 0.33 | 1.67 | 0.33 |
| | AA (2/255) | 98.47 | 54.93 | 57.28 | 44.33 | 44.33 | 44.33 | 41.00 | 92.28 | 90.00 | 6.33 | 1.00 | 6.33 | 1.00 |
| | AA (1/255) | 100.0 | 48.17 | 51.97 | 54.00 | 54.00 | 54.00 | 47.33 | 82.66 | 80.58 | 15.00 | 6.67 | 15.00 | 6.67 |
| | AA (0.5/255) | 100.0 | 48.54 | 52.46 | 53.00 | 53.00 | 52.19 | 44.31 | 70.53 | 71.17 | 25.00 | 14.00 | 24.62 | 13.79 |
| **CelebaHQ32_4** | AA (8/255) | 100.0 | 69.49 | 74.25 | 31.67 | 21.67 | 31.67 | 21.67 | 87.79 | 88.71 | 11.33 | 9.67 | 11.33 | 9.67 |
| | AA (4/255) | 99.43 | 56.20 | 58.90 | 43.33 | 37.67 | 43.08 | 37.46 | 72.07 | 71.14 | 27.33 | 29.33 | 27.17 | 29.16 |
| | AA (2/255) | 68.26 | 51.86 | 50.43 | 49.00 | 50.67 | 33.45 | 34.59 | 59.31 | 56.24 | 40.00 | 46.67 | 27.30 | 31.86 |
| | AA (1/255) | 27.70 | 45.34 | 46.29 | 57.82 | 55.44 | 16.02 | 15.36 | 49.82 | 51.26 | 52.38 | 47.96 | 14.51 | 13.28 |
| | AA (0.5/255) | 10.91 | 54.69 | 45.45 | 40.17 | 57.26 | 4.38 | 6.25 | 53.44 | 44.75 | 43.59 | 58.12 | 4.76 | 6.34 |
| **CelebaHQ64_4** | AA (8/255) | 100.0 | 84.26 | 86.90 | 14.33 | 2.67 | 14.33 | 2.67 | 97.17 | 100.0 | 2.67 | 0.00 | 2.67 | 0.00 |
| | AA (4/255) | 100.0 | 64.23 | 58.35 | 35.67 | 40.00 | 35.67 | 40.00 | 90.88 | 94.86 | 10.33 | 4.67 | 10.33 | 4.67 |
| | AA (2/255) | 99.31 | 55.19 | 52.60 | 43.33 | 46.00 | 43.03 | 45.68 | 72.51 | 73.61 | 28.33 | 31.67 | 28.13 | 31.45 |
| | AA (1/255) | 69.94 | 48.59 | 51.09 | 54.00 | 49.00 | 37.77 | 34.27 | 55.30 | 57.63 | 47.00 | 43.33 | 32.87 | 30.31 |
| | AA (0.5/255) | 28.14 | 48.36 | 48.45 | 53.33 | 53.00 | 15.01 | 14.91 | 52.68 | 48.04 | 46.00 | 55.00 | 12.94 | 15.48 |
| **CelebaHQ128_4** | AA (8/255) | 100.0 | 71.52 | 72.76 | 24.67 | 23.00 | 24.67 | 23.00 | 94.21 | 99.17 | 5.00 | 0.00 | 5.00 | 0.00 |
| | AA (4/255) | 100.0 | 93.57 | 92.88 | 3.00 | 0.00 | 3.00 | 0.00 | 98.67 | 100.0 | 1.33 | 0.00 | 1.33 | 0.00 |
| | AA (2/255) | 100.0 | 54.94 | 48.26 | 45.33 | 53.67 | 45.33 | 53.67 | 82.99 | 89.07 | 18.67 | 7.67 | 18.67 | 7.67 |
| | AA (1/255) | 98.02 | 51.51 | 47.08 | 48.67 | 54.33 | 47.71 | 53.25 | 63.18 | 60.17 | 37.67 | 41.33 | 36.92 | 40.51 |
| | AA (0.5/255) | 61.98 | 50.74 | 48.52 | 48.67 | 53.67 | 30.17 | 33.26 | 53.22 | 53.36 | 47.67 | 47.00 | 29.55 | 29.13 |

**Appendix** 2: Different datasets are attacked by *AutoAttack* but with a different epsilons for the perturbation. The ASR falls for different datasets.