# OpenReview forum: "Is RobustBench/AutoAttack a suitable Benchmark for Adversarial Robustness?"
_AAAI.org/2022/Workshop/AdvML — AAAI-22 AdvML Workshop ShortPaper_

### Official Review · Reviewer_kxJ3 · 2021-11-27
**Review of paper "Is AutoAttack/AutoBench a suitable Benchmark for Adversarial Robustness?"**

**Rating:** 7
**Confidence:** 4

**Review:**

This paper discusses several limitations of AutoAttack and RobustBench for robustness evaluation. It points out two limitations: 1) The adversarial examples generated by AutoAttack can be easily detected. 2) AutoAttack does not generalize well to datasets with higher resolutions. This paper provides excessive experiments to validate these two findings.

Overall, it is interesting to see such discussions on AutoAttack, which is the most popular benchmark to evaluate adversarial robustness. The findings are insightful and could be useful for future research. The authors are encouraged to further complete their paper to a long version.

---

### Decision · Program_Chairs · 2021-12-01

**Decision:**

Accept (Short Paper)

**Comment:**

The reviewer agrees to accept this paper.